# Novel Adaption of the SARC-F Score to Classify Pediatric Hemato-Oncology Patients with Functional Sarcopenia

**DOI:** 10.3390/cancers15010320

**Published:** 2023-01-03

**Authors:** Emma J. Verwaaijen, Patrick van der Torre, Josef Vormoor, Rob Pieters, Marta Fiocco, Annelies Hartman, Marry M. van den Heuvel-Eibrink

**Affiliations:** 1Princess Máxima Center for Pediatric Oncology, 3584CS Utrecht, The Netherlands; 2Wolfson Childhood Cancer Research Centre, Northern Institute for Cancer Research, Newcastle University, Newcastle upon Tyne NE1 7RU, UK; 3Utrecht Cancer Center, University Medical Center Utrecht, 3584CX Utrecht, The Netherlands; 4Mathematical Institute, Leiden University, 2333CA Leiden, The Netherlands; 5Department of Biomedical Data Science, Section Medical Statistics, Leiden University Medical Center, 2333ZA Leiden, The Netherlands; 6Department of Pediatric Physiotherapy, Erasmus Medical Center-Sophia Children’s Hospital, 3015CN Rotterdam, The Netherlands; 7Division of Child Health, Wilhelmina Children’s Hospital, 3584EA Utrecht, The Netherlands

**Keywords:** sarcopenia, pediatric oncology, leukemia, muscle weakness, screening, muscle mass

## Abstract

**Simple Summary:**

Children treated for hemato-oncological diseases are at risk of muscle deterioration, such as loss of muscle mass and muscle weakness, with consequent impaired physical functioning. An easy screening tool will facilitate the early identification of children at risk and will support clinical decision making. We investigated the accuracy of an easy screening tool: the pediatric SARC-F (PED-SARC-F), for identifying functional sarcopenia in pediatric hemato-oncology patients. Functional sarcopenia indicates low muscle strength combined with low physical performance. We showed that the PED-SARC-F has a 90% accuracy in identifying pediatric hemato-oncology patients with functional sarcopenia. A PED-SARC-F cut-off point of ≥5 had the highest specificity (91%) and limits unnecessary assessments in patients who are not at risk of sarcopenia. This tool can be used to identify children that need a physiotherapy assessment and further interventions to prevent physical deterioration during and shortly after treatment for a hemato-oncology disease.

**Abstract:**

Sarcopenia in pediatric hemato-oncology patients is undesirable because of the consequences it may have for treatment continuation and outcome, physical abilities and participation in daily life. An easy-to-use screening tool for sarcopenia will facilitate the identification of children at risk who need interventions to prevent serious physical deterioration. In the elderly, the use of the SARC-F score as a case-finding tool for sarcopenia is recommended. The aim of this cross-sectional study was to investigate the accuracy of the pediatric SARC-F (PED-SARC-F) for identifying sarcopenia in pediatric hemato-oncology patients, including the determination of a cut-off point for clinical use. Patients 3–20 years of age, under active treatment or within 12 months after treatment cessation were eligible. Patients had a physiotherapy assessment including a PED-SARC-F (0–10) and measurements of muscle strength (handheld dynamometry), physical performance (various tests) and/or muscle mass (bio-impedance analysis), as part of the standard of care. Spearman’s correlation coefficient (r_s_) between the PED-SARC-F and physiotherapy outcomes were calculated. Structural sarcopenia was defined as low appendicular skeletal muscle mass (ASMM) in combination with low muscle strength and/or low physical performance. Functional sarcopenia indicated low muscle strength combined with low physical performance. Multiple logistic regression models were estimated to study the associations between the PED-SARC-F and structural/functional sarcopenia. To evaluate which cut-off point provides the most accurate classification, the area under the receiver operating characteristic curve (AUCs), sensitivity and specificity per point were calculated. In total, 215 assessments were included, 62% were performed in boys and the median age was 12.9 years (interquartile range: 8.5–15.8). The PED-SARC-F scores correlated moderately with the measurements of muscle strength (r_s_ = −0.37 to −0.47, *p* < 0.001) and physical performance (r_s_ = −0.45 to −0.66, *p* < 0.001), and weakly with ASMM (r_s_ = −0.27, *p* < 0.001). The PED-SARC-F had an AUC of 0.90 (95% confidence interval (CI) = 0.84–0.95) for functional sarcopenia and 0.79 (95% CI = 0.68–0.90) for structural sarcopenia. A cut-off point of ≥5 had the highest specificity of 96% and a sensitivity of 74%. In conclusion, we adapted the SARC-F to a pediatric version, confirmed its excellent diagnostic accuracy for identifying functional sarcopenia and defined a clinically useful cut-off point in pediatric hemato-oncology patients.

## 1. Introduction

Treatment of children with hemato-oncological diseases is often intensive and associated with symptom-burdened trajectories which may involve (prolonged) hospitalizations. These children are generally immunocompromised, at higher risk of infections, and prone to malnutrition, general malaise and immobilization. In particular, the administration of glucocorticoids and vincristine contributes to muscle deterioration [1,2] and peripheral neuropathy [3,4], which can aggravate immobilization and consequent loss of muscle mass and strength. This all can lead to seriously impaired physical performance with negative consequences for quality of life.

The combination of decreased muscle strength and muscle mass loss is referred to as sarcopenia: a generalized muscle deficiency [5,6]. Sarcopenia has been associated with increased adverse health outcomes and mortality in adults with various diseases [6,7]. In pediatric cancer patients, sarcopenia is a relatively understudied condition, and its prevalence, causal factors and consequences have not been entirely elucidated [8]. However, studies in children with acute lymphoblastic leukemia (ALL) indicate the necessity of awareness of sarcopenia during therapy [9,10,11]. Muscle mass loss during ALL therapy was associated with the number and duration of hospital admissions [9], occurrence of invasive fungal infections [10] and even with impaired survival [11].

The European Working Group on Sarcopenia recommends the use of the SARC-F as a case-finding tool for sarcopenia [6]. The SARC-F is a quick self-report score including five questions addressing muscular strength, the ability to walk, rise from a chair, climb stairs and the experience of falls [12]. The SARC-F has been shown to be a valid and consistent instrument for detecting sarcopenia in the elderly [13] and in adult cancer settings [14]. Previous meta-analyses showed that the SARC-F had low-to-moderate sensitivity and a high specificity for identifying older adults at risk of sarcopenia [15,16]. A cut-off point of a score ≥ 4 has been proposed to detect probable sarcopenia and has been associated with poorer health outcomes [13].

Based on the results in the elderly population, we implemented a slightly adapted version as part of our clinical physiotherapy care for children with hemato-oncological diseases. However, to our knowledge, the clinical usefulness of the SARC-F had never been investigated in pediatric populations. Therefore, the aim of this project was to determine the accuracy of the pediatric SARC-F (PED-SARC-F) in our (national) single-center pediatric hemato-oncology cohort, using physiotherapy outcome measures. Subsequently, we determined a clinically useful cut-off score of the PED-SARC-F to easily classify children with sarcopenia.

## 2. Materials and Methods

### 2.1. Study Design and Patients

In this cross-sectional study, patients between 3 and 20 years of age, under active treatment or within 12 months after the cessation of treatment at the hemato-oncology and stem cell department of the Princess Máxima Center for Pediatric Oncology, Utrecht, the Netherlands, were included. Only patients that had given informed consent to use their data for research purposes were considered; this was approved by the Medical Ethical Committee (MEC-2016-739).

The PED-SARC-F has been implemented as part of the standard physiotherapy assessment since December 2018. Data from assessments performed between that time and January 2021 were retrieved from the electronic patient records and anonymized. Patients had to have an in-patient physiotherapy consultation including the administration of the PED-SARC-F, with at least one assessment of muscle strength, physical performance or muscle mass to be included. If a patient had undergone more than one physiotherapy consultation, these could only be included if the assessments happened during different treatment phases (e.g., during intensive chemotherapy and after therapy cessation). Patients with known (motor)developmental or neurological disorders, as well as symptomatic osteonecrosis or bone fractures that resulted in impaired physical functioning, were excluded.

### 2.2. PED-SARC-F

The SARC-F is a short screening tool that encompasses five self-reported questions, which reflect physical changes associated with sarcopenia [13]. For the particular usage in our pediatric oncology population, we slightly adapted the text of the original questions, resulting in a pediatric version: the PED-SARC-F (Appendix A). Parents and/or patients were asked to estimate the difficulties they had observed over the last 2 weeks for each of the four items: ‘lifting something heavy’, ‘walking’, ‘rising from the floor’ and ‘climbing stairs’, as follows: 0 = no difficulties, 1 = some difficulties and 2 = a lot of difficulties or unable to perform. The fifth item ‘number of falls’ was scored as 0 = zero falls, 1 = 1–3 falls and 2 = ≥4 falls. Total scores range from 0–10. Cognitive debriefing was not performed prior to administration.

### 2.3. Physiotherapy Assessment

The standard physiotherapy assessment consisted of measuring muscle strength, physical performance and muscle mass. Not all measurements could be performed in every patient. As the assessment took place as part of clinical care, it depended on the fitness of the patient and the assessment being a burden.

A total of three different measures of muscle strength with handheld dynamometry (HHD) were performed. Handgrip strength was assessed in a sitting position with the elbow unsupported and flexed at 90° using a Jamar HHD (Sammons Preston, Bolingbrook, IL, USA). Hip flexion and knee extension strength were measured with the eccentric break technique protocol in the standardized positions [17] using the MicroFET-2 HHD (Hoggan Health Industries, Salt Lake City, UT, USA). With the break technique, the physiotherapist applies the force needed to overpower the patient who is extending the knee or flexing the hip, thereby eliciting an eccentric contraction from the patient. All HHD measurements were carried out bilaterally, the mean of three repeats was compared to normative values and Z-scores were calculated [17,18].

Aspects of physical performance were assessed with three different tests. First, the 10 m walk test (10-MWT) was used [19]. The child was asked to walk a marked distance of 10 m at normal pace, independently, without using a support. The fastest time in seconds of three tries was scored. Second, the Time to Rise from the Floor test (TRF) [19] was carried out. The child was asked to get up as fast as possible from sitting in a cross-legged position on the floor. Per protocol [19], this test was performed twice, the fastest performance in seconds was scored. Third, the Timed Up and Down Stairs (TUDS) test was performed [20]. The time required to ascend and descend a flight of stairs (10 steps) was measured in seconds.

Muscle mass was measured using bioimpedance analysis (BIA) (Tanita MC-780, Tanita Corporation, Tokyo, Japan). The measurement procedure required the child to stand barefoot on the analyzer and hold a handgrip on each side for approximately 10 s. Appendicular skeletal muscle mass (ASMM) was calculated with correction for light indoor clothing. As reference data for Dutch children were unavailable, to estimate Z-scores, we used age and sex-specific mean and standard deviation values from a UK population (5–18 years), acquired using the same Tanita software [21]. Due to a lack of BIA reference values of 3–4-year-old children, we used sex and age-specific expected values of ASMM (kilogram), derived by a dual-energy X-ray absorptiometry prediction equation in Canadian children [22].

### 2.4. Structural and Functional Sarcopenia Definition

Two definitions for sarcopenia were used. Structural sarcopenia encompassed a low ASMM in combination with low muscle strength and/or low physical performance. Functional sarcopenia indicated low muscle strength combined with low physical performance (Figure 1). Patients could meet the criteria for both structural and functional sarcopenia when they had a low ASMM and low muscle strength and low physical performance, indicating that impairments were prevalent on a structural and functional level.

To define ‘low’, we selected the patients with values of the lowest 20% of the cohort, due to a lack of pediatric reference values for sarcopenia, which is in line with previous studies using functional outcome measures of frailty [23,24]. Low muscle strength was defined as a (for sex and age) standardized score of handgrip strength, hip flexion strength or knee extension strength in the lowest 20% of all measured patients. If a child was physically incapable, i.e., movement against gravity or resistance was limited, this was also classified as low muscle strength.

Low physical performance was defined as a score in the highest 20% (higher score equals slower performance) of the 10-MWT, TRF or TUDS. One of the measurements had to be low for a child to be labeled as having low physical performance. If a child was incapable of rising or (stair)walking independently, this was also classified as low.

Low ASMM was defined as a (for sex and age) standardized score in the lowest 20% of all measured patients.

### 2.5. Statistical Analyses

All data were expressed as means and standard deviations (SDs) for normally distributed variables or median and interquartile ranges (IQRs) for skewed distributions and number (percent) for categorical variables.

We performed Spearman’s rank correlation analyses to assess the correlations between the PED-SARC-F and the objectively measured components: handgrip strength, hip flexion strength, knee extension strength, 10-MWT, TRF, TUDS and ASMM. A Spearman coefficient (r_s_) ranges from −1 to +1, where an r_s_ of 0 to 0.3 (0 to −0.3) means a negligible correlation, 0.3 to 0.5 (−0.3 to −0.5) means a low correlation, 0.5 to 0.7 (−0.5 to −0.7) means a moderate correlation, 0.7 to 0.9 (−0.7 to −0.9) means a high correlation, and 0.9 to 1.0 (−0.9 to −1.0) describes a perfect correlation [25].

Multiple logistic regression analyses were used to assess the associations between the PED-SARC-F and the binary outcomes of functional and structural sarcopenia. Age was used in both models, while sex was incorporated only in the model for functional sarcopenia due to the small sample size for structural sarcopenia. The results are presented as odds ratio (OR) along with 95% confidence intervals (CI). We computed cluster robust standard errors for the estimated parameters to deal with multiple assessments. The area under the receiver operating characteristic curve (AUC) was estimated to assess the discriminative accuracy of the PED-SARC-F. The AUC values lie between 0.5–1; 0.5 suggests no discrimination and >0.9 = outstanding discrimination [26].

Subsequently, we examined which PED-SARC-F cut-off point (0–10) had the highest diagnostic accuracy for detecting functional sarcopenia. For clinical purposes, we have determined the cut-off point assembling the highest AUC and the highest specificity as the most clinically relevant, yielding a low number of false positives. We therefore calculated the AUC, sensitivity, specificity, positive predictive value (PPV) and negative predictive value (NPV) [27] for each PED-SARC-F score, to be able to distinguish between the number of false positive and false negative cases per cut-off point.

All analyses were performed in the RStudio environment Version 1.4.1106 for Windows [28].

## 3. Results

### 3.1. Patients

In total, 215 physiotherapy assessments in 167 patients were included in this study. Among the 167 patients, 126 had a single assessment, 34 children had two and 7 children had an assessment at three different time points. Sarcopenia may occur with different severity and during different phases of treatment. In the case of more than one assessment per patient it always took place at different treatment phases, as specified in Appendix A. The characteristics of the 167 individual patients at their first physiotherapy assessment are depicted in Appendix A. Due to the cross-sectional design of this study, we considered each physiotherapy assessment (during a different treatment phase) as an individual assessment in our analyses, rendering 215 assessments.

The majority of the 215 assessments was performed in boys (62.3%), the median age was 12.9 years (IQR: 8.5–15.8), the median height Z-scores were −0.27 (IQR: −0.8–1.2), the weight Z-scores were 0.13 (IQR: −0.8–1.2) and the BMI Z-scores were 0.33 (IQR: −0.8–1.6) (Table 1).

### 3.2. PED-SARC-F Scores and Physiotherapy Assessment

The median PED-SARC-F score was 2 (range: 0–10). The results corresponding to the individual questions are depicted in Appendix A. The mean Z-score of handgrip strength (n = 100) was −0.75, SD: 1.0. The mean Z-score of hip flexion strength (n = 123) was −2.3 SD: 1.2, and of knee-extension strength (n = 139) was −1.5, SD: 1.2. The median walking pace in the 10-MWT (n = 144) was 1.23 m per second (IQR: 1–1.4). The median TRF time (n = 141) was 2 s (IQR: 1.4–3). The mean ASMM Z-score (n = 150) was −0.6 (IQR: −1.2–0.0). The results of the total physiotherapy assessments, including the number of children that were not able to perform a test, are specified in more detail in Table 2. The results of the sub-cohort of the 167 individual patients on their first physiotherapy assessment were not different from the results of the complete cohort (*p* > 0.05). 

### 3.3. Correlations of PED-SARC-F with Muscle Strength, Physical Performance and ASMM

The PED-SARC-F scores correlated with handgrip strength (r_s_ = −0.37, *p* < 0.001), knee extension strength (r_s_ = −0.34, *p* < 0.001) and hip flexion strength (r_s_ = −0.47, *p* < 0.001), i.e., higher PED-SARC-F scores correlated weakly to decreased Z-scores (Figure 1A). The PED-SARC-F scores correlated with 10-MWT (r_s_ = 0.45, *p* < 0.001), TRF (r_s_ = 0.66, *p* < 0.001) and TUDS (r_s_ = −0.64, *p* < 0.001), i.e., a higher PED-SARC-F score correlated moderately to slower performance (Figure 1B). A lower PED-SARC-F correlated weakly with lower ASMM Z-scores (r_s_ = −0.27, *p* < 0.001) (Figure 1C).

### 3.4. Diagnostic Accuracy of the PED-SARC-F for Structural and Functional Sarcopenia

In total, 16 patients met the criteria for structural sarcopenia (7.4%) and 46 (21.4%) for functional sarcopenia (Figure 2). The classification of patients with structural, functional or no sarcopenia per the PED-SARC-F score is illustrated in Figure 3. Logistic regression analyses showed that the odds for functional sarcopenia were two times higher for every PED-SARC-F point increase (OR = 2.07, 95%CI: 1.68–2.55), in models adjusted for sex and age. The model provided an AUC equal to 0.90 (95%CI: 0.84–0.95), indicating that 90% of the children with functional sarcopenia are classified correctly by the PED-SARC-F (Table 3). The odds of structural sarcopenia were 1.5 points higher for every PED-SARC-F score increase; the AUC was equal to 0.79 (95%CI: 0.687–0.90) (Table 3).

In addition, the diagnostic accuracy of functional sarcopenia for different cut-off points of the PED-SARC-F was calculated. A score of ≥5 provides the best diagnostic accuracy (AUC: 0.82, 95%CI: 0.74–0.91), with a specificity of 91%, sensitivity of 74%, PPV of 68% and an NPV of 93%. Robust standard errors provided similar accuracy as the models estimated with traditional standard errors.

The results for each PED-SARC-F score are described in Appendix A.

## 4. Discussion

Sarcopenia in children receiving intensive treatment for hemato-oncology diseases is undesirable because of the consequences it may have for treatment continuation, physical abilities, motor and neurodevelopment and participation in daily life.

In the current study, we evaluated the diagnostic accuracy of our novel pediatric version of the SARC-F (PED-SARC-F), which has been used as part of our clinical physiotherapy care for pediatric hemato-oncology patients.

To the best of our knowledge, this is the first attempt to evaluate the use of the SARC-F in a pediatric population. For clinical purposes, we extended the definition of sarcopenia by separating structural and functional sarcopenia. So far, the sequence of events leading to sarcopenia in children with cancer has not yet been clearly discerned. In childhood cancer survivors, muscle mass can decrease prior to strength and function loss being observed, which is different from the elderly, who experience decreases in muscle strength preceding muscle mass loss [29]. The results of our study suggest that low muscle strength (28%) and impaired physical performance (32%) were more prevalent than low muscle mass (14%). This may indicate that muscle strength can deteriorate in the absence of or with only minimal muscle mass loss, which could be explained by the fact that muscle strength depends on many complex physiological mechanisms. Alternatively, loss of muscle strength and performance may be the early signs of sarcopenia in this population. Furthermore, some children may partially recover their muscle mass, but muscle quality (and thus performance) may stay compromised. The precise understanding between the occurrence of muscle mass loss and strength or function loss in children treated for hemato-oncological diseases has yet to be elucidated. The results of this study show that the PED-SARC-F is a suitable and accurate cross-sectional screening tool to identify pediatric hemato-oncology patients with functional sarcopenia (diagnostic accuracy: 90%, 95%CI: 0.84–0.95). Although the PED-SARC-F correlated only moderately with physical performance outcomes and even had a low correlation with muscle strength measurements, it is excellent at identifying patients who suffer from low muscle strength and impaired performance, after adjustment for sex and age.

We found a weak correlation between the PED-SARC-F and ASMM. In contrast, we found a moderate-to-good accuracy for the detection of structural sarcopenia (discriminative ability: 79%, 95%CI: 0.68–0.9). This may be explained by the fact that a relatively large number of children with low ASMM also had impaired strength or performance (n = 10). The weak correlation may be diminished due to the fact that BIA could not be performed in bedridden patients, who probably had lower muscle mass but who did not have an ASMM measurement. Moreover, there is uncertainty about the reliability of BIA in children with high fat percentages [30], and also hydration status may affect the measurements, as it causes an increase in the body’s electrical resistance [31]. Both overweight and disturbed fluid balance can occur in hemato-oncology patients, thus this may have influenced the results. Unfortunately, in our clinical cohort we had no availability of muscle mass measures other than BIA. Imaging techniques are frequently used in pediatric research but for clinical care these methods have several limitations. Computed tomography (CT) is undesirable due to radiation exposure and the calculation of muscle mass on CT scans is time-consuming. The current gold standard is magnetic resonance imaging (MRI), which is expensive, poorly accessible, time-consuming and needs mobilization of the patient to a radiology department and sedation in younger children. Another reliable technique is dual-energy X-ray absorptiometry (DXA), which has the same disadvantages as MRI, making these techniques unsuitable for routine evaluations. 

However, the finding that the PED-SARC-F is less sensitive for detecting low muscle mass, resembles the results of adult studies concerning community-dwelling elderly [32], patients with chronic kidney disease [33] and cancer patients [34], which also showed that the SARC-F is better in detecting alterations in muscle strength and function rather than muscle mass deterioration.

In previous studies in the elderly, a SARC-F cut-off of ≥4 points was proposed to identify patients with probable sarcopenia [13,35]. Our analyses show that a cut-off point of ≥5 for the PED-SARC-F had the highest AUC and specificity for detecting functional sarcopenia in pediatric hemato-oncology patients. We aimed for the highest specificity because in clinical care, such a cut-off point reduces the number of false positives and therefore limits unnecessary assessments in patients that are not at risk.

Some methodological limitations should be addressed. First, patients in this study were selected for the availability of their physiotherapy assessments (selection bias). This may have led to stronger associations because the patients were more likely to have muscle weakness or physical impairments. Secondly, there is no uniform definition and there are no cut-off points with reference values for sarcopenia in children yet. We therefore decided for a margin of 20% in our definition, in line with previous studies using functional outcome measures of frailty [23,24].

Based on the results of this study, we recommend the use of the quick and easily self-reported PED-SARC-F as a screening instrument by pediatric oncologists or nurse specialists, to identify hemato-oncology patients at risk of functional impairments due to loss of muscle strength. In patients with a PED-SARC-F score of ≥5, referral to a (pediatric) physiotherapist for specific function assessment and interventions should be considered.

Current evidence for interventions to improve muscle strength in children with cancer is not very comprehensive. It is known that treatment-induced denervation of muscle fibers and mitochondrial dysfunction may occur, but the direct molecular impact of cancer treatment is unknown. It has been suggested that exercise may potentially stimulate repair/replacement of damaged mitochondria [36]. A number of trials have been performed demonstrating that exercise is safe and feasible, even during intensive treatment [37,38], but the effectiveness of interventions during cancer treatment on muscle mass and strength in pediatric patients has not been shown yet. For future research, it is important to determine the adequate moment for prevention and training during treatment, and to determine the most beneficial training for the individual patient, yielding the most resilient results. Further knowledge of the biological mechanism behind muscle dysfunctions in these patients is needed to develop successful interventions. The PED-SARC-F may play an important role in selecting patients who are at risk and may be a valuable tool in longitudinal evaluations.

## 5. Conclusions

To conclude, we adapted the SARC-F questionnaire to the PED-SARC-F, an easy-to-use self-reporting tool, and showed its value for identifying functional sarcopenia in pediatric hemato-oncology patients. We recommend a PED-SARC-F cut-off score of ≥5 as clinically useful. This tool can identify children that may need a physiotherapy assessment and further interventions to prevent physical deterioration during and shortly after treatment for a hemato-oncology disease.

## Figures and Tables

**Figure 1 cancers-15-00320-f001:**
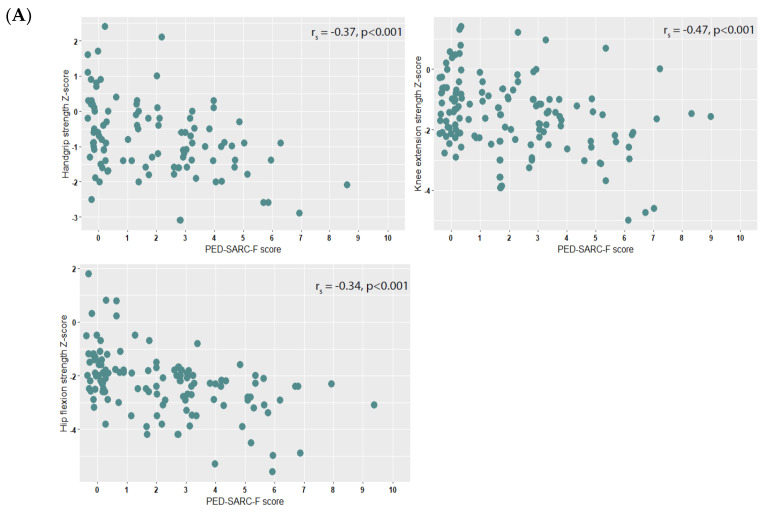
(**A**). Scattergraphs showing muscle strength measures by PED-SARC-F score; (**B**). Scattergraphs showing physical performance tests by PED-SARC-F score; (**C**). Scattergraph showing appendicular skeletal muscle mass (ASMM) by PED-SARC-F score. Abbreviations: r_s_ = Spearman’s correlation coefficient.

**Figure 2 cancers-15-00320-f002:**
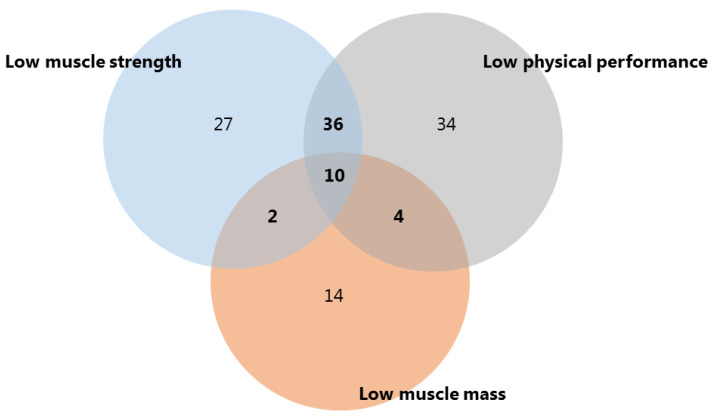
The occurrence of the sarcopenia components in the assessments. Co-occurrence of the components led to a number of 16 (2 + 10 + 4) for structural sarcopenia and 46 (36 + 10) assessments meeting the criteria for functional sarcopenia.

**Figure 3 cancers-15-00320-f003:**
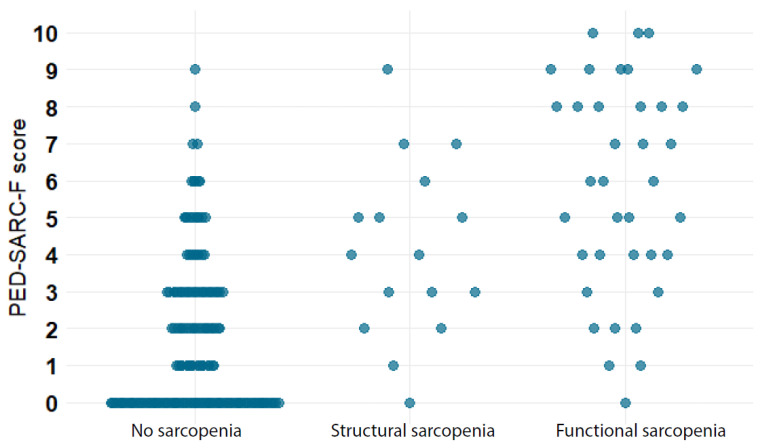
Scattergraph: No, structural and functional sarcopenia by PED-SARC-F score.

**Table 1 cancers-15-00320-t001:** Patient characteristics at physiotherapy assessments (n = 215).

	No.	%
Sex		
Boy	134	62.3
Girl	81	37.7
Type of hematological disease		
Acute lymphoblastic leukemia	129	60
Acute myeloid leukemia	24	11.2
Chronic myeloid leukemia	6	2.8
Hodgkin lymphoma	8	3.7
Non-Hodgkin lymphoma	16	7.4
Myelodysplastic syndrome	8	3.7
Fanconi anemia	14	6.5
Aplastic anemia	3	1.4
Other *	7	3.3
Treatment phase		
Intensive chemotherapy	54	25.1
Maintenance chemotherapy	57	26.5
1–12 months after chemotherapy cessation	19	8.9
Pre SCT conditioning phase	31	14.4
3–12 months post SCT	54	25.1
Assessment performed during		
Clinical admission	36	16.7
Daycare admission/Outpatient clinic visit	179	83.3
Body Mass Index, categories		
Underweight	18	8.4
Normal Weight	141	65.6
Overweight	41	19
Obesity	15	7
	Mean	Median [IQR]
Age, years	12.1	12.9 [8.5 to 15.8]
Height, SDS	−0.31	−0.27 [−0.8 to 1.2]
Weight, SDS	0.25	0.13 [−0.8 to 1.2]
Body Mass Index, SDS	0.33	0.33 [−0.8 to 1.6]

Abbreviations: SCT = stem cell transplantation, IQR = interquartile range, SDS = standard deviation score. * Blastic plasma cytoid dendritic cell neoplasm (n = 2), common variable immunodeficiency (n = 1), de novo acute promyelocytic leukemia (n = 1), Diamond–Blackfan anemia (n = 1), Langerhans cell histiocytosis (n = 1), Paroxysmal nocturnal hemoglobinuria (n = 1).

**Table 2 cancers-15-00320-t002:** Results of physiotherapy assessments (n = 215).

	N	Median	Interquartile Range	Measurement Not Performed	
				Incapable	Other ^I^
Muscle strength measurements					
Handgrip strength	100			14	101
Dominant hand, kilograms		17.4	8.7 to 27.5		
Dominant hand, Z-score ^II^		−0.75	1.0		
Hip flexion, strength	123			23	69
Left hip, Newton		126	101 to 192		
Right hip, Newton		133	98 to 177		
Mean Left + Right hip, Z-score ^II^		−2.3	1.2		
Knee extension strength	139			19	57
Left leg, Newton		184	131 to 259		
Right leg, Newton		186	128 to 269		
Mean Left + Right leg, Z-score ^II^		−1.5	1.2		
Physical performance measurements					
10 m Walk Test	144			11	60
Time, seconds		8.1	7.4 to 9.5		
Meters per second		1.23	1.0 to 1.4		
Time To Rise from the Floor	141	2	1.4 to 2.9	26	48
Time, seconds					
Timed Up and Down Stairs	115			27	103
Time, seconds		7.8	5.8 to 12.4		
Step per second		0.4	0.3 to 0.6		
Muscle mass measurement					
Bio-electrical impedance analysis	175			NR	70
ASMM, kg		14.5	8.4 to 19.8		
ASMM, % ^II^		28.4	25.4 to 30.7		
ASMM, Z-score ^II^		−0.60	−1.2 to 0.0		

Abbreviations: ASMM = Appendicular Skeletal Muscle Mass, NR = Not reported. ^I^ Measurement was not performed because of various reasons, i.e., hospital admission or connected to intravenous line, parent or child refusal, too much of a burden, child was fatigued/nauseous, had poor understanding or was too young. ^II^ Presented as mean and standard deviation.

**Table 3 cancers-15-00320-t003:** The predictive accuracy of the PED-SARC-F for detecting functional and structural sarcopenia.

	OR	95% CI	AUC (95% CI)
**Functional sarcopenia**			0.90 (0.84 to 0.95)
PED-SARC-F, per point	2.07	1.68 to 2.55	
Sex, boy vs. girl	0.71	0.28 to 1.78	
Age, years	0.98	0.89 to 1.08	
**Structural sarcopenia**			0.79 (0.68 to 0.90)
PED-SARC-F, per point	1.54	1.23 to 1.93	
Age, years	0.99	0.87 to 1.12	

Abbreviations: OR = odds Ratio, CI = confidence interval, AUC = area under the curve.

## Data Availability

The data presented in this study are available on request from the corresponding author. The data are not publicly available due to the privacy restrictions of the study participants.

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
