# Peer review of "Novel Adaption of the SARC-F Score to Classify Pediatric Hemato-Oncology Patients with Functional Sarcopenia"

_cancers, 2023, doi:10.3390/cancers15010320_

Round 1

Reviewer 1 Report

The present paper adapts a tool for sarcopenia assessment in elderly (SARC-F) to a pediatric population, showing its usefulness and reliability in detecting children with low muscle function, who would benefit from physical therapy before/during/after treatment of cancer. I believe the PED-SARC-F may represent a useful, important tool for clinicians and I thank the author for their work.

I have some comments attached. I hope the authors will find those comments useful to improve the manuscript quality. 

Methods: could you briefly describe the eccentric break-technique protocol? Potentially, in a similar manner to what you do later for physical performance test, in only 2-3 rows.

I was wondering if another possible method to score low muscle mass/function/strength might be the following: obtaining the average and standard deviation for each parameter in the sample, and considering children with low muscle mass/function/strength those children whose value are below 2 standard deviations from the average. Have the authors considered this approach? Does this approach provide similar results to the one used in the present work?

Results: I can’t find supplemental figure 2.

I do understand that the design of the study is cross-sectional. However, I don’t believe it is correct to pool the repeated measures obtained at different time point in the same child as if they derived from different children. Perhaps you could perform 2 differential analyses, one including 167 patients (cross-sectional) and the second focusing on the predictive power of SARC-F in children during the time-course of their admission-treatment-post operation.

Another possibility may be to pool (if you have enough n) the different time points (chemotherapy, after chemotherapy, SCT, post SCT) and consider the power of SARC-F in children at different phases of treatment, also focusing on whether this tool is more predictive of sarcopenia in children at admission, during chemotherapy or SCT. Perhaps you can add this analysis for a more in-depth interpretation of your results.

Discussion:

L289-291: please remove repetition of the sentence.

I believe that, whenever a human experiences a loss of muscle mass, this will necessarily be followed by a certain degree of muscle strength loss. Conversely, muscle strength can decrement (and it has been shown to do so) also in the absence or minimal muscle mass loss, as muscle strength depends on many complex electro-physiological mechanisms. Thus I don’t believe we can think of these two phenomena as “independent from one another” but rather the degree to which we observe muscle atrophy and weakness can be uncorrelated, as you happened to see in this study. Maybe you can be more precise about this in lines 303-305.

Reviewer 2 Report

This study focused on to use of SARC-F in sarcopenia patients, especially in children. SARC-F is a well-known method to understand muscle strength and deterioration in elderly patients with multiple diseases. The authors claim that this is the first study which shows the use of SARC-F in pediatric patients of acute lymphoblastic leukaemia. They have claimed that the sensitivity and accuracy of this method will improve the understanding of the risk of sarcopenia in young patients and will help to reduce unnecessary assessment in patients who are not at risk of sarcopenia.

1. As authors indicated that low muscle strength and impaired physical performance were more prevalent than low muscle mass. Does this claim have any correlation with the patient’s weight?

2. Study shows that muscle strength measurements are associated with handgrip strength, hip flexion and knee extension strength. Did they find any conflict in the sex and age difference, i.e., one particular age/sex is more tolerant or sensitive to specific measurements? The data shows the cohort study, but individual data is missing in physiotherapy assessments.

3 Muscle tissue changes considerably with age at the celular level - PMID: 33536212. Please dicuss this in the introduction section.

Round 2

Reviewer 2 Report

Authors have adrressed my comments. I have no further comments.